# Pharmacological Properties and Function of the PxOctβ3 Octopamine Receptor in *Plutella xylostella* (L.)

**DOI:** 10.3390/insects13080735

**Published:** 2022-08-16

**Authors:** Hang Zhu, Zheming Liu, Haihao Ma, Wei Zheng, Jia Liu, Yong Zhou, Yilong Man, Xiaoao Zhou, Aiping Zeng

**Affiliations:** 1College of Plant Protection, Hunan Agricultural University, Changsha 410128, China; 2Institute of Agricultural Biotechnology, Hunan Academy of Agricultural Sciences, Changsha 410125, China

**Keywords:** *Plutella xylostella*, octopamine receptor, pharmacology, gene expression, RNAi

## Abstract

**Simple Summary:**

The diamondback moth (*Plutella xylostella*) is a global migratory pest that primarily destroys cruciferous vegetables. Due to the extensive application of insecticides, *P. xylostella* has developed resistance to more than 90 types of active insecticidal ingredients, and this has exacerbated its control. Octopamine plays a key role in the neural activity of invertebrates because it regulates various physiological functions and behaviors in insects. The actions of octopamine have been shown to be mediated via the activation of octopamine receptors (OARs). In this study, we identified an OAR gene *(PxOctβ3*) in *P. xylostella*, analyzed its molecular characteristics and patterns of expression, and expressed it in HEK-293 cells to not only analyze the pharmacological properties of the encoded PxOctβ3 receptor but also identify its specific agonists and antagonists. RNA interference-mediated inhibition of the expression of PxOctβ3 resulted in shorter durations of pupation in the diamondback moth. These results indicate that PxOctβ3 plays an important role in the physiology of *P. xylostella*.

**Abstract:**

The diamondback moth (*Plutella xylostella*) is one of the most destructive lepidopteran pests of cruciferous vegetables, and insights into regulation of its physiological processes contribute towards the development of new pesticides against it. Thus, we investigated the regulatory functions of its β-adrenergic-like octopamine receptor (PxOctβ3). The open reading frame (ORF) of *PxOctβ3* was phylogenetically analyzed, and the levels of expression of the receptor mRNA were determined. This ORF was also cloned and expressed in HEK-293 cells. A series of octopamine receptor agonists and antagonists were tested against PxOctβ3. We showed that the receptor is a member of the Octβ3 protein family, and an analysis using quantitative PCR showed that it was expressed at all developmental stages of *P. xylostella*. Octopamine activated PxOctβ3, resulting in increased levels of intracellular cAMP. Furthermore, the agonists naphazoline, clonidine, 2-phenethylamine, and amitraz activated the PxOctβ3 receptor, and naphazoline was the most effective. Only metoclopramide and mianserin had significant antagonistic effects on PxOctβ3, whereas yohimbine, phentolamine, and chlorpromazine lacked obvious antagonistic effects. The injection of double-stranded RNA in an RNA interference assay indicated that PxOctβ3 regulates development in *P. xylostella*. This study demonstrated the pharmacological properties and functions of PxOctβ3 in *P. xylostella*, thus, providing a theoretical basis for the design of pesticides that target octopamine receptors.

## 1. Introduction

The lepidopteran *Plutella xylostella* is a worldwide migratory pest that is primarily controlled using chemical pesticides [1]. However, this has led to its evolution of resistance to 95 insecticides, including all the major types of pesticides, making *P. xylostella* one of the pests of global crops that is the most resistant [2]. Therefore, it is imperative to develop new insecticides that are both highly selective to target compounds and have low toxicity to nontarget organisms [3]. In this study, new insights for the control of *P. xylostella* were generated by determining the molecular regulatory mechanisms and pharmacological properties of a β_3_-adrenergic-like octopamine receptor (PxOctβ3) in *P. xylostella.*

Biogenic amines act as neurotransmitters, neuromodulators, and neurohormones in insects, as well as other protostomes, which mediate their effects by binding to specific receptors [4,5]. Biogenic amine receptors are types of G-protein-coupled receptors (GPCRs) that have seven transmembrane structures and transmit signals through secondary messengers after activation by the corresponding biogenic amines [6]. Octopamine (OA) is a multifunctional biogenic amine found at high concentrations only in invertebrates [7]. Octopamine plays a vital role in a wide range of physiological processes, including fighting, motivation, aggression, muscle regulation and performance, ovulation, glycogen decomposition, fat metabolism, desensitization, learning and memory, and the circadian rhythm [8,9,10,11].

Octopamine acts by binding to specific octopamine receptors (OARs), which are putative target molecules for new insecticides [12]. Based on molecular and pharmacological data, insect octopamine receptors can be further grouped as α- or β-adrenergic-like octopamine receptors [7,13,14]. The α-adrenergic-like octopamine receptors induce the release of Ca^2+^ from intracellular stores (α_1_) or inhibit adenylyl cyclases (α_2_), whereas the β-adrenergic-like receptors stimulate adenylyl cyclases and thereby increase the levels of [cAMP]I [15,16,17,18]. Studies have shown that insects generally have multiple β-adrenergic-like octopamine receptors [19,20]. For example, three additional genes in *Drosophila* (DmOctβ1 to DmOctβ3) have been characterized that encode β-adrenergic-like octopamine receptors that generate cAMP signals [21,22]. Honeybees (*Apis mellifera*) possess four β-adrenergic-like octopamine receptors (AmOctβ1 to AmOctβ4) [23]. Octβ2, the dominant subject of current research on insect OA receptors, controls ovulation in *D. melanogaster* and is highly expressed in the fertilized female reproductive tract [24]. Similarly, both the injection of OAR antagonists and interference of N1Octβ2 in the brown planthopper (*Nilaparvata lugens*) resulted in a decrease in egg production [25]. The octopamine receptor antagonists mianserin and phentolamine affected motility in adult rice stem borers (*Chilo suppressalis*) by inhibiting CsOctβ2 [18]. Similarly, the activation of specific OA receptors in the skeletal and cardiac muscles of *Drosophila* was required for exercise adaptation, and the expression of Octβ2 in skeletal muscles improved endurance and speed [26]. Conversely, apart from the finding that the expression of Octβ3 in the prothoracic gland of *Drosophila* is required for development [27], there is a dearth of knowledge on the functions of Octβ3. In this study, we determined the function and pharmacological properties of an Octβ3 receptor of *P. xylostella*. PxOctβ3 was activated by OA, which resulted in increased cellular levels of cAMP. It was highly agonized by naphazoline and antagonized by metoclopramide and mianserin. PxOctβ3 regulated the rates of pupation in *P. xylostella.*

## 2. Materials and Methods

### 2.1. Insects

*P. xylostella* larvae were originally purchased from Henan Jiyuan Baiyun Industrial Co., Ltd., Henan Province in 2018 and reared at 25 ± 1 °C and 65 ± 5% relative humidity with 16 h/8 h light/dark. Newly hatched larvae were fed on radish (*Raphanus sativus*) seedlings. Second-instar larvae were fed on cabbage (*Brassica oleracea* var. capitata) leaves, and the adults were fed on 5% honey water.

### 2.2. Total RNA Extraction and cDNA Synthesis

*P. xylostella* at different growth stages, including eggs, 1–4 instar larvae, prepupal and pupal, male and female adults, were obtained in triplicate. TRIzol (Invitrogen, Carlsbad, CA, USA) reagent (0.8 mL) was added to each of the samples, which were then homogenized and stored at −80 °C.

Fourth-instar larvae were dissected under a light microscope, and the following tissues were sampled: head, epidermis, midgut, and Malpighian duct from 10, 5, 5, and 10 larvae, respectively. This was repeated three times. The samples were then lysed with 0.5 mL of RNA extraction reagent and stored at −80 °C for later use.

Total RNA was extracted with RNA extraction kits (RNA-easy Isolation Reagent (Vazyme Biotech, Nanjing, China)) according to the manufacturer’s instructions and quantified with a Nanodrop analyzer 1000 (Thermo Fisher Scientific, Waltham, MA, USA). A script Q RT Supermix for qPCR (+gDNAwiper) Kit (Vazyme) protocol was used for reverse transcription. One μg of total RNA in a 20 μL reaction was used to synthesize cDNA, which was then stored at −80 °C for later use.

### 2.3. Cloning of the Full-Length PxOctβ3 ORF Sequence

Homologous sequence alignment by ClustalW2 suggested that the predictive gene sequence XM_011557721.1 in the *P. xylostella* transcription database may be the putative gene for *PxOctβ3*. Hence, XM_011557721.1 was cloned and used for further analysis. To amplify the complete open reading frame (ORF) of *PxOctβ3*, specific primers (OctβR_XhoI_F and OctβR_PstI_R) (Table 1) were designed for PCR amplification. Phanta Super Fidelity DNA polymerase (Vazyme) was used to ensure high-fidelity PCR amplification. The cycling conditions were as follows: initial denaturation at 95 °C for 3 min, followed by 30 cycles of 95 °C for 15 s, 64 °C for 20 s, and 72 °C for 2 min. The final extension was performed at 72 °C for 5 min. The PCR products were visualized using 1.0% agarose gel electrophoresis and then purified before they were cloned into a pEGFP-N1 vector in between *Xho*I and *Pst*I restriction enzyme sites. *Escherichia coli* DH5α competent cells were then transformed with the ligation product (pEGFP-N1-PxOctβ3). LB (Luria Bertani) agar plates supplemented with 50 μg/mL kanamycin were used to select for positive clones. Plasmids were then extracted and successful recombination events were verified by sequencing.

### 2.4. Bioinformatic Analysis of the PxOctβ3

Pairwise alignments of *PxOctβ3* ORF were performed in BLAST (https://blast.ncbi.nlm.nih.gov/Blast.cgi, accessed on 10 October 2019). Multiple sequence alignments were performed using the ClustalX2 algorithm (https://www.ebi.ac.uk/Tools/msa/clustalo/, accessed on 10 October 2019) and edited with Jalview software (version 2.11, http://www.jalview.org/, accessed on 10 October 2019). The protein sequence analytical tools provided by the ExPASy website were used to analyze the transmembrane region of the protein (TMHMM 2.0, http://www.cbs.dtu.dk/services/TMHMM-2.0/, accessed on 10 October 2019), the PKC phosphorylation sites (GPS5.0, http://gps.biocuckoo.org/, accessed on 10 October 2019), the potential N-glycosylation sites (NetNGLyc 1.0 Server, http://www.cbs.dtu.dk/services/NetNGlyc/, accessed on 10 October 2019), and the conserved cysteine palmitoylation sites (GPS-Lipid 1.0, http://lipid.biocuckoo.org/, accessed on 10 October 2019). To study the evolution of the PxOctβ3 protein, the amino acid sequences of octopamine receptors from different insects were downloaded from GenBank and used as inputs in MEGA 7.0 (http://www.megasoftware.net/, accessed on 10 October 2019) using the neighbor-joining method and bootstrapped with 1000 replications.

### 2.5. Analysis of the Pattern of Expression of PxOctβ3

The level of expression of *PxOctβ3* at different growth stages and tissues of fourth-instar *P**. xylostella* larvae were quantitatively analyzed by fluorescence with ribosomal protein L32 as the internal reference gene. The total RNA was reverse-transcribed to cDNA, which was amplified by sequence-specific primers (Octβ3_qF, Octβ3_qR) and (Rpl32_qF, Rpl32_qR) (Table 1). This process was conducted in triplicate for each sample. A Roche Diagnostic LightCycler^®^96 PCR instrument (Indianapolis, IN, USA) was used for quantitative fluorescence analysis. The cycle conditions were as follows: initial denaturation at 95 °C for 30 s, followed by 40 cycles of denaturation at 95 °C for 10 s and annealing at 60 °C for 30 s. The fluorescence was measured at the end of each cycle. The amplification efficiencies of *PxOctβ3* and Rpl32 were 97.6% and 96.8%, respectively. The results from the three replicates were analyzed, and differences in the levels of expression between the samples were compared using the 2^−ΔΔCT^ method.

### 2.6. Construction of a Cell Line That Stably Expresses PxOctβ3

HEK-293 cells were cultured at 37 °C and 5% CO_2_ concentration in D-MEM medium (HyClone Laboratories, Inc., Logan, UT, USA) supplemented with 10% fetal bovine serum (Gibco, Thermo Fisher Scientific), penicillin (100 U/mL) (Gibco), and streptomycin (0.1 mg/mL) (Gibco). Six-well cell culture plates (NEST; Wuxi, China) were inoculated with the cultivated HEK-293T cells at a cell density of 0.8 × 10^6^ per well one day before the cells were transfected. Transfections were conducted using a Lipofectamine^®^2000 Transfection Reagent Kit (Invitrogen) following the manufacturer’s instructions. HEK-293 cells in their exponential phase were transformed by PcDNA3.1-PxOctβ3 (transfected) and pcDNA3.1-GFP (control) plasmids (4 μg DNA/well). After transfection, the cells were digested with trypsin, and 6-well cell culture plates were inoculated with transfected cells. These cells were cultured for 48 h; then, G418 (800 µg/mL) was used to select for positive transformants. Those that stably expressed PxOctβ3 were selected for characterization and cAMP assays.

### 2.7. Cyclic AMP (cAMP) Assays

The cells were treated with octopamine, tyramine, dopamine, histamine, and serotonin at a final concentration of 10 μM, as well as with the agonists naphazoline, clonidine, 2-phenylethylamine, and amitraz. A volume of 100 μM yohimbine, mianserin, phentolamine, and other antagonists were mixed with 10 nM of octopamine to act on the cells. An analysis of the drug delivery processes was performed as previously described [12,28]. Cell samples were collected, and the concentrations of cAMP were measured using a cAMP Assay Kit (R&D Systems, Minneapolis, MN, USA) according to the manufacturer’s instructions. A BCA (bicinchoninic acid) Protein Assay kit was used to determine the protein concentration of the cell lysate. The ratio of cAMP (pmol/mL) to protein concentration (mg/mL) in the cell lysate was used to compare the concentrations of cAMP in the samples collected. A series of concentrations (1 nM–100 μM) of octopamine was used to stimulate the cells transfected with PxOctβ3 to evaluate the effect of its activation. The data were processed using a special dose–response, allosteric EC_50_ shift, nonlinear regression model in GraphPad PRISM software version 8.0 (San Diego, CA, USA). The EC_50_ was defined as the concentration of agonist that produced a response that was half of the maximum amount.

### 2.8. Synthesis and Microinjection of dsRNA

Specific primer pairs (Table 1) that flanked the T7 promoter sequences at their 5′ ends were designed and used to amplify fragments of the *PxOctβ3* open reading frame (ORF) that had been selected as the double-stranded RNA (dsRNA) sequence for the RNA interference (RNAi) experiments. These primers targeted an intergenic gene-specific region to avoid potential off-target effects. An in silico analysis in BLASTN (https://www.ncbi.nlm.nih.gov/, accessed on 10 October 2019) showed that they did not bind to any other OA receptor genes, which further confirmed the specificity of the selected dsRNA fragment. DsRNA was generated using a T7 RNAi Transcription Kit (Vazyme). The concentration and quality of dsRNA were determined by a NanoDrop spectrometer 2000 (Thermo Fisher Scientific) and agarose gel electrophoresis, respectively.

One-day-old fourth-instar larvae of *Pl**. xylostella* were injected with dsRNA using a Nanoject II Auto-Nanoliter Injector (Drummond Scientific Co., Broomall, PA, USA) microinjector. A total of 200 ng/larva of dsRNA was injected. The larvae were then reared at 25 °C under the conditions previously described and fed an artificial diet. The silencing efficiency was assessed at 24 h, 48 h, and 72 h by assessing the differences in pupation rates. In total, 100 larvae (n = 5, N = 20) were assessed in the dsOctβ3 treatment group and 100 larvae (n = 5, N = 20) were assessed in the double-stranded green fluorescent protein dsGFP) treatment control group. Each treatment contained 20 biological replicates (N = 20).

### 2.9. Data Analysis

All the studies were replicated three to five times and plotted as the mean ± standard error using GraphPad PRISM^®^ software version 8.0 (GraphPad Software, Inc., San Diego, CA, USA). The statistical significance of differences between the samples and between treatments was assessed using two-tailed, nonpaired *t*-tests at an alpha level of 0.05. One-way analyses of variance (ANOVA) paired with Tukey–Kramer posthoc tests were used to test for the differences among multiple groups of data.

## 3. Results

### 3.1. Molecular Cloning and Sequence Analysis

The sequencing results confirmed the successful cloning of the putative *PxOctβ3* ORF. It was 1425 bp and encoded a protein of 474 aa, whose predicted molecular weight and isoelectric point were 52,554 Da and 8.49, respectively. Moreover, the *PxOctβ3* protein was predicted to contain seven transmembrane domains, which would each be 23 aa in length. The amino acid sequence of *PxOctβ3* was 41.05% similar with that of *DmOctβ3*, 53.91% with *TcOctβ3*, and 30.82% with *AmOctβ3*. Multiple sequence alignments identified two conserved cysteines whose loci were in between TM2 and TM3, and TM4 and TM5, which were in the extracellular loops. The second phenylalanine after the FxxxWxP motif in TM6 and that after the NPxxY motif in TM7 were also conserved. These conserved cysteine and phenylalanine sites are not only necessary for aminergic GPCRs but are also conserved in all previously reported adrenergic receptors [29] (Figure 1).

The amino acid sequences of 28 OARs were downloaded from GenBank, and the sequences of their transmembrane regions were used for phylogenetic reconstruction. The putative *PxOctβ3* protein clustered in an Octβ3Rs clade that contained *AmOctβ3*, *DmOctβ3*, and *TcOctβ3* (Figure 2).

### 3.2. Pattern of Expression of PxOctβ3

To understand the profile of expression of the *PxOctβ3* receptor gene in *P. xylostella*, quantitative PCR (qPCR) was used to quantify the level of expression of *PxOctβ3* at different growth stages. *PxOctβ3* was expressed at all the growth stages, but it was expressed the least in second-instar larvae. The levels of expression increased significantly after the prepupa stage. The levels of expression in female and male adults differed significantly in that the female adults had lower levels, which were equivalent to those of the larval stage, whereas male adults had the highest levels (Figure 3A).

The levels of expression of the PxOctβ3 in the head, epidermis, midgut, and Malpighian tubules of the fourth-instar larvae were also determined. The levels of expression of PxOctβ3 in different tissues of the *P. xylostella* larvae were not significantly different. However, the levels of expression were lower in the head and epidermis and higher in the midgut and Malpighian tubules (Figure 3B).

### 3.3. Pharmacological Property Analysis of PxOctβ3

Insect Octβ3Rs are specifically activated by either octopamine or tyramine, resulting in increased levels of intracellular cAMP [30,31]. Thus, corresponding experiments were conducted on transformed HEK293 cells that stably expressed PxOAB3.

HEK-293 cells that expressed the putative PxOctβ3 protein were stimulated by octopamine, tyramine, and dopamine, which resulted in a significant increase in the intracellular concentration of cAMP. The intracellular cAMP concentrations after octopamine, tyramine, and dopamine treatment were 11.7-fold, 9.6-fold, and 2.3-fold of those before treatment, respectively. Treatment with other biogenic amines did not significantly change the concentrations of intracellular cAMP. Moreover, these results corroborated the successful cloning of the octopamine receptor PxOctβ3 of *P. xylostella* (Figure 4A).

The cells were treated with different concentrations of octopamine (1 Nm–100 μM) to further investigate the effect of octopamine concentration on recombinant HEK-293 cells that overexpressed the putative PxOctβ3 (Figure 4B). The EC_50_ value of octopamine was approximately 65 nM.

There are various octopamine receptor agonists and antagonists. Thus, we tested the effects of selected octopamine receptor agonists and antagonists on PxOctβ3. Metoclopramide and mianserin significantly inhibited the increase in cAMP concentrations induced by octopamine. Mianserin had the highest antagonistic effect, followed by metoclopramide, then chlorpromazine, and the least effective was phentolamine, whose antagonistic effect was not significant (Figure 4C). A total of 10 μM of agonists, including naphazoline, clonidine, phentolamine, and amitraz, were used to treat HEK-293 cells that expressed PxOctβ3. The PxOctβ3 receptors were activated and the concentrations of cellular cAMP increased. Naphazoline had the strongest stimulatory effect and increased the concentration of intracellular cAMP by 9.1-fold, whereas clonidine, amitraz, and 2-phenylethylamine were barely effective agonists (Figure 4D).

### 3.4. RNA Interference and Pupation Rate Determination after RNAi

To evaluate the effect of downregulating the expression of *PxOctβ3* gene in *P. xylostella*, we performed specific dsRNA injection experiments on the fourth-instar 1-day-old larvae. Compared with the control group (injection of dsGFP), injection with dsRNA significantly decreased the level of expression of *PxOctβ3* after 48 h (61.9%) and 72 h (37.3%). This showed that the selected dsRNA fragment effectively interfered with the expression of the *PxOctβ3* gene, and this interference was the most effective after 48 h (Figure 5A). Monitoring the phenotypic changes after injection indicated that the survival rate of *P. xylostella* had no significant difference with the control group, but the pupation time of the larvae in the treatment group was shortened after interference with the expression of *PxOctβ3*(Figure 5B).

## 4. Discussion

GPCRs play important roles in insects; thus, the disruption or overactivation of certain GPCRs causes developmental inhibition, reduced reproductive capacity, or even death. This makes them potential targets for insecticides [32,33,34]. The β-adrenergic-like octopamine receptor (Octβ) is a target for formamidine insecticides. Since the first β-adrenergic-like octopamine receptor gene was cloned from *D. melanogaster*, octopamine receptor genes from insects, including *Apis mellifera*, *Bombyx mori*, *Bactrocera dorsalis*, *Chilo suppressalis*, *Periplaneta americana*, *Plutella xylostella*, and *Nilaparvata lugens*, have been similarly successfully cloned [18,21,25,28,35,36]. However, these studies were primarily focused on Octβ2. DmOctβ2 is an important target of amitraz, and *Drosophila* with DmOctβ2 knockouts was unresponsive to amitraz. In contrast, the artificial activation of DmOctβ2 neurons induced amitraz-like poisoning symptoms [37]. Three β-adrenergic-like octopamine receptors (Octβ1, Octβ2, and Octβ3) are found in *D**. melanogaster*, *Trichoplusia ni*, *Pieris rapae*, and *Nicrophorus vespilloides* among others. However, there is a dearth of knowledge on the functions of Octβ3. Indeed, whether Octβ3 plays an important role in *P. xylostella* or is a target of amitraz remains unknown. In this study, we cloned a PxOctβ3 cDNA sequence of *Pl. xylostella.* PxOctβ3 had high sequence similarities with members of the Octβ3 receptor family and formed a phylogenetic clade with DmOctβ3, AmOctβ3, and TcOctβ3. OA and high concentrations of TA activated the receptor that results in an increase in the intracellular concentration of cAMP in HEK-293 cells that express PxOctβ3. These results corroborated the successful cloning and subsequent expression of the *PxOctβ3* gene.

HEK293 cells are widely used in the study of insect OARs [10,18,38]; thus, HEK-293 cells were used in this study to stably express PxOctβ3. Several different biogenic amines, agonists, and antagonists were tested to explore the pharmacological properties of PxOctβ3. PxOctβ3 reacted the most strongly to octopamine but also reacted to tyramine. This interaction could be because tyramine is the biosynthetic precursor of octopamine, and these two phentolamines have highly similar structures. In addition, the response of PxOctβ3 to dopamine was significant (*p* ≤ 0.05), but the activation effect was much lower than that of octopamine. The effective medium concentration EC_50_ of octopamine for PxOctβ3 was 6.50 × 10^−8^ M, which was similar to that of Octβ1 [30] but lower than that of OA2B2 [28]. PxOctβ3 was more sensitive to octopamine than TcOctβ3 (EC_50_ of 8.68 × 10^−7^ M) but less sensitive than DmOctβ3 (EC_50_ of 1.40 × 10^−8^ M) [39,40].

Among the four agonists, naphazoline had the highest activation ability followed by phenylethylamine, then clonidine, and lastly amitraz. These results were similar to those from studies on DmOctβ3 in *Drosophila* [40]. Only mianserin and metoclopramide strongly inhibited the cAMP response caused by OA in the antagonist assay, and this corroborated a study of β-adrenergic-like octopamine receptors in *Drosophila* and *Apis mellifera* [23], where both metoclopramide and mianserin showed strong antagonistic effects. Thus, metoclopramide and mianserin are potential broad-spectrum antagonists for insect pharmacology and behavioral research. Yohimbine, a specific antagonist of tyramine receptors, had no significant antagonistic effect on PxOctβ3; this further confirmed PxOctβ3 as an octopamine receptor. Phentolamine is a typical antagonist of α-adrenergic-like receptors and had a strong antagonistic effect on DmOctβ2 in *Drosophila* [21] and CsOctβ2 in *C. suppressalis* [18]. However, no significant antagonistic effect was observed with PxOctβ3. This could be due to the specificity of different biological receptor structures. Moreover, amitraz is a formamidine acaricide that primarily controls mites and induces its miticidal activity by either directly or indirectly acting on OAR [41,42,43]. We compared the activation effects of OA and amitraz on the β-adrenergic-like receptors of *P. xylostella*, including PxOctβ1, PxOctβ2, and PxOctβ3. The activation ability of OA to the three respective β-adrenergic-like receptors was 2.78-, 1.46-, and 3.75-fold higher than that of amitraz at concentrations of 10 µM. This suggests that PxOctβ2 is the main target of amitraz, corroborating the conclusion that DmOctβ2 could be the only target of amitraz in *D. melanogaster*. Therefore, it is worthwhile to investigate the physiological function of Octβ3 and explore its potential as a target for insecticides.

Tissue localization analysis provided a key to understanding the function of OARs in *P. xylostella*. A previous study pointed out that *DmOctβ3* is obviously expressed in the central nervous system of Drosophila, particularly in the mushroom body [40]. The level of expression of *TcOctβ3* is dominant in the central nervous system. Our results revealed that the *PxOctβ3* was expressed in the epidermis, midgut, Malpighian tubules, and head, but the overall difference was not significant. It is worth noting that *PxOctβ3* was not highly expressed in the head. This was probably the case because our study used the whole head of the *P. xylostella* as the experimental material, and the nerve tissue was not separated and detected. Another result shows that the level of expression of the gene increased several times during the stages from larva to adult, and the level of expression in the male adults was significantly higher than that of the females. However, *TcOctβ3* has been found to be highly expressed in early larvae, and there may be differences in the level of expression of this gene among different species. In addition, we found that the level of expression of *PxOctβ3* differed in females and males, which was also reflected in *PxOctβ1* and *PxOctβ2* [28,29,30]. In addition, this study confirmed that an increase in the nerve impulse responses of male pheromone-sensitive receptor neurons (RNs) after octopamine injection in silkworm (*Bombyx mori*) is either absent or less sensitive in females [44]. This could be the reason for the differential expression of OARs between males and females. Therefore, PxOctβ1, PxOctβ2, and PxOctβ3 may have similar functions or synergistic effects, which forms a reference for further research on the physiological function of PxOctβ3.

Currently, the research on how octopamine receptor subtypes regulate insect physiological activities is focused on the β_2_-octopamine receptor, which may be involved in the regulation of reproductive behaviors, such as mating and spawning in insects [25,45]. There have been few studies on the β_3_-octopamine receptor in lepidopteran insects. *Drosophila* is subject to the autocrine regulation of ecdysone synthesis by a β_3_-octopamine receptor in the prothoracic gland [27]. In this study, the function of PxOctβ3 was explored by RNAi assays. Pupation of the *P. xylostella* larvae was accelerated compared with the control group after interfering with the expression of PxOctβ3. This effect differed from the delayed molting effect observed after interference with the expression of Octβ3 in *Drosophila*. This difference could be due to either the difference in species or the use of a Gal4-upstream activation sequence (UAS) system to specifically interfere with Octβ3 expression in the prothoracic gland of *Drosophila*. RNAi was not performed on the prothoracic gland of *P. xylostella* larva in this study due to a lack of sufficient tools for genetic manipulation. In addition, whether this phenotype after RNAi is related to the high expression of *PxOctβ3* in the pupal stage also merits further confirmation. However, experiments on RNAi with Lepidoptera have frequently been found to be difficult to achieve [46]. Thus, we intend to knock out the receptor using a CRISPR/Cas9 technique to study the role of PxOctβ3 in the physiological activity of *P. xylostella* in more detail.

## 5. Conclusions

This study cloned a candidate OA receptor gene (*PxOctβ3*) from *P. xylostella*. Analyses of evolutionary relationships and alignments of its deduced protein sequence were performed, and the temporal and spatial patterns of expression of the OA receptor gene in *P. xylostella* were identified. The pharmacological properties of the *PxOctβ3* gene were analyzed in HEK-293 cells that expressed its protein product. The physiological function of PxOctβ3 in *P. xylostella* was explored by RNAi. These findings advance studies on the physiological functions of PxOctβ3.

## Figures and Tables

**Figure 1 insects-13-00735-f001:**
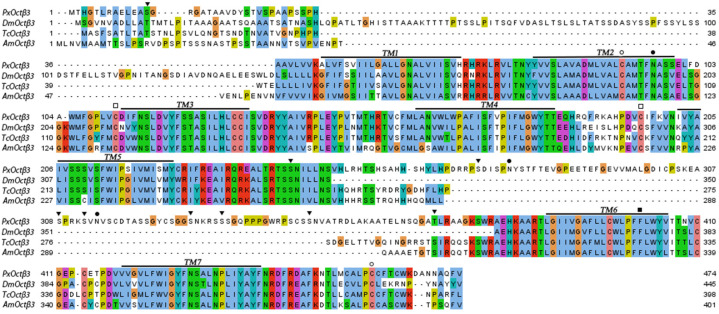
Amino acid sequence alignment of PxOctβ3 and orthologous receptors from *Apis mellifera* (XP_006557730.1), *Drosophila melanogaster* (NP_001034043.2), and *Tribolium castaneum* (NP_001280505.1). The seven predicted transmembrane regions are denoted by TM1to M7. Potential N-glycosylation and phosphorylation sites by PKC are marked with filled circles and triangles, respectively. Potential palmitoylation sites are marked with empty circles. Two conserved cysteine residues are marked with empty squares. The second phenylalanine after the FxxxWxP motif in TM6, a unique feature of aminergic GPCRs, is marked with a filled square. GPCR, G-protein-coupled receptors; PKC, protein kinase C.

**Figure 2 insects-13-00735-f002:**
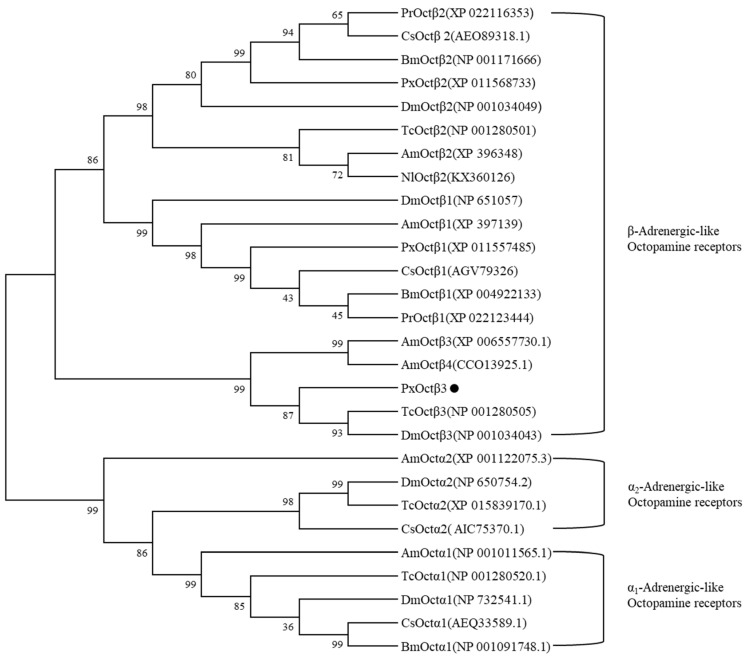
Phylogenetic tree analysis of representative members of the different OARs. Phylogenetic reconstruction was performed using the amino acid positions that were found in the TM1 to 7 regions. Numbers in the middle of the branches denote percentage bootstrap support with 1000 replications per branch. The receptor names followed by their GenBank accession numbers are listed in the tree. Bm, *Bombyx mori*; Pr, *Pieris rapae*; Cs, *Chillo suppressalis*; Px, *Plutella xylostella*; Dm, *Drosophila melanogaster*; Am, *Apis mellifera*; Tc, *Tribolium castaneum*; Ni, *Nilaparvata lugens*; OAR, octopamine receptor.

**Figure 3 insects-13-00735-f003:**
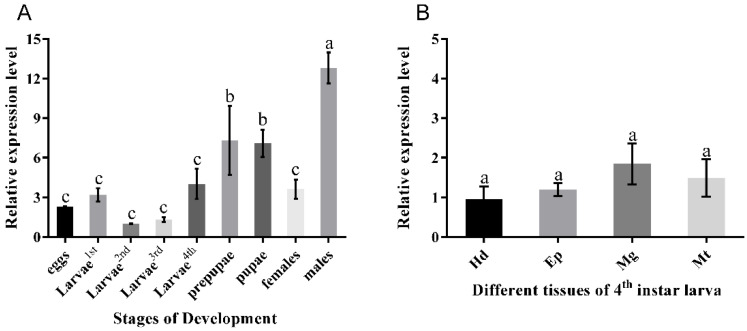
Relative level of expression (mean ± SE) of Octβ3 in *Plutella xylostella*. (**A**) The levels of expression of *PxOctβ3* genes in different development stages (first to fourth larvae, prepupae, pupae, adults, and eggs). The level of expression of *PxOctβ3* in the second-instar larvae was used for comparison. (**B**) The levels of expression of *PxOctβ3* in the tissues (head (Hd), epidermis (Ep), midgut (Mg), and Malpighian tubules (Mt)) of the fourth-instar larvae. *PxOctβ3* in the Hd was used for comparison. The data were the mean values of three independent experiments (n = 3) and normalized to the endogenous ribosomal protein L32, which served as the internal control. Different letters above the bars indicate significant differences in different stages and tissues (*p* < 0.05). Statistical comparisons were performed using one-way analysis of variance (ANOVA) followed by Tukey’s honest significant difference (HSD) test to determine whether significant differences occurred.

**Figure 4 insects-13-00735-f004:**
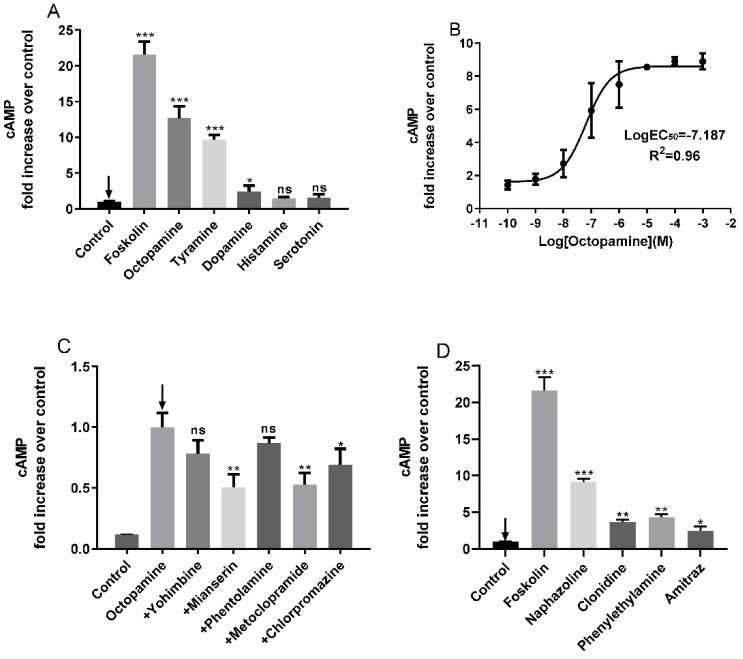
Effects of various biogenic amines, agonists, and antagonists on the concentration of cAMP in HEK-293 cells that overexpressed the *PxOctβ3* gene. The pcDNA3.1-PxOctβ3 plasmid was stably expressed in HEK-293 cells, which were used to detect changes in the levels of intracellular cAMP after treatment with different biogenic amines, agonists, and antagonists. The relative concentration of intracellular cAMP was calculated by defining the concentration of cAMP in the control cells as 1 (4.08 ± 0.39 pmol/mg protein). (**A**) Effects of various biogenic amines on the levels of cAMP in HEK-293 cells that stably expressed the *PxOctβ3* gene. HEK-293 cells that stably expressed PxOctβ3 were treated with forskolin as a positive control. The relative change in [cAMP] was provided as a multiple of the value obtained for the untreated control (=1). (**B**) Effects of different doses of octopamine on the levels of cAMP in HEK-293 cells that stably expressed PxOctβ3. (**C**) The effects of different antagonists (100 μM) on the levels of cAMP in HEK-293 cells that stably expressed PxOctβ3. Data represent the mean of three experiments. (**D**) Effects of various agonists on the levels of cAMP in HEK-293 cells that stably expressed the *PxOctβ3* gene. HEK-293 cells that stably expressed PxOctβ3 were treated with forskolin as a positive control. The statistical analysis was based on a one-way analysis of variance (ANOVA) followed by Tukey’s multiple comparison test. *** *p* ≤ 0.001, ** *p* ≤ 0.01, * *p* ≤ 0.05, ns means no significant.

**Figure 5 insects-13-00735-f005:**
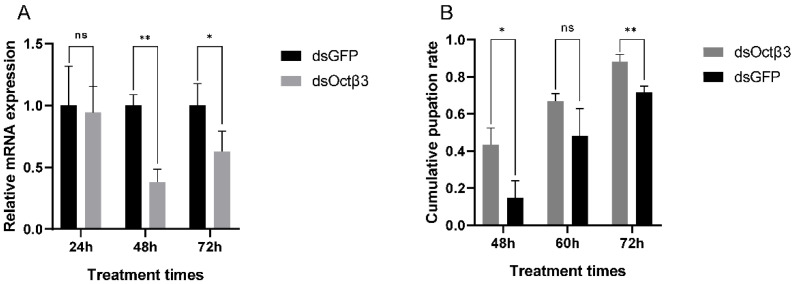
Effect of silencing PxOctβ3 expression on pupation in *Plutella xylostella.* (**A**) The relative levels of mRNA expression of PxOctβ3 after injection with 200 ng dsOA2B3 and dsGFP. DsRNAs were injected into one-day-old fourth-instar larvae. The relative levels of mRNA expression of PxOctβ3 were determined using RT-qPCR at 24 h, 48 h, and 72 h postinjection. (**B**) The pupation rate of *P. xylostella* larvae at 48 h, 60 h, and 72 h postinjection. Error bars represent the standard error of the calculated means based on three biological replicates. Asterisks indicate significant differences (Student’s *t*-test, *p* < 0.05). dsGFP, double-stranded green fluorescence protein; dsRNA, double-stranded RNA; RT-qPCR, quantitative reverse transcription PCR. ** *p* ≤ 0.01, * *p* ≤ 0.05, ns means no significant.

**Table 1 insects-13-00735-t001:** Primers used in this study.

Primer Name	Primer Sequences	Purpose of Primers
Octβ3R_XhoI_F	CTCGAGCCACCATGACCCATGGCACGCTGC	DNA cloning
Octβ3R_PstI_R	TCTCTATATACTGCAGCACAAACTGTGC	
Octβ3_qF	CGTCTAGCGTATCGTTTTGGAT	Quantitative RT-PCR
Octβ3_qR	GTGGAGGTAGTGTGAATGGTGT	
Rpl32_qF	TGCCCAACATTGGTTACGG	
Rpl32_qR	ACGATGGCCTTGCGCTTC	
dsOctβ3_F	TAATACGACTCACTATAGGGCCATCGTCAGACCCTTAG	RNAi
dsOctβ3_R	TAATACGACTCACTATAGGCGTGCTGTAGTTCGGACTTAT	
Octβ3_ygF	CTGCCCTTCTTCCTGTGG	
Octβ3_ygR	GTGCGTTATTAGCGTCCTT	

## Data Availability

The data presented in this study are available in the article.

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
