# Peer review of "Pharmacological Properties and Function of the PxOctβ3 Octopamine Receptor in Plutella xylostella (L.)"

_insects, 2022, doi:10.3390/insects13080735_

Round 1
Reviewer 1 Report
General comments
This is a short paper on the characterization of an octopamine receptor in the diamondback moth Plutella xylostella which is an important pest insect. The PyOA2B3 receptor was heterologously expressed in HEK293 cells and its pharmacological profile was established. In addition the authors measured the mRNA expression level in different developmental stages and tissues. RNAi experiments were performed to down-regulate receptor expression in order to find possible functions of the receptor protein. In my opinion, the experiments have been carried out correctly and provide further results on octopaminergic signaling in insects. In contrast, the introduction and the discussion of the present work in relation to the overall area and the selection of references are somewhat idiosyncratic in places. In addition, the authors have severe problems with their English writing. The manuscripts contains numerous grammatical and stylistic errors, which make it difficult to follow in places. I strongly recommend that authors get help from a native speaker.
Specific comments
1. Generally, the octopamine receptor nomenclature of Evans & Maqueira (2005) should be used: Octopamine receptors can be differentiated in α-adrenergic-like and β-adrenergic-like receptors. The α-adrenergic-like receptors evoke Ca2+ release from intracellular stores (α1) or inhibit adenylyl cyclases (α2) whereas the β-adrenergic-like receptors stimulate adenylyl cyclases and thereby raise [cAMP]i. The classification into OA1 and OA2 receptors has gone out of fashion (see e.g. the following recent reviews: Ozoe 2021; Blenau and Baumann, 2016; Ohta and Ozoe, 2014).
2. Line 62 “OARs are divided into two”: Did you mean two receptor subfamilies? There are actually three subfamilies of specific octopamine receptors: α1, α2 and β (see Specific comment 1). The recent discovery of the α2-adrenergic octopamine receptors (e.g., Wu et al., 2014; Qi et al., 2017; Blenau et al., 2020; Nakagawa et al., 2022) is neglected throughout the manuscript.
3. Lines 65-66 “β-octopamine receptors are subdivided into OA2B1, OA2B2, and OA2B3 receptors”: The number of subtypes of β-adrenergic-like octopamine receptors present depends on the insect species. For example, honey bees possess four β-adrenergic-like octopamine receptors (Balfanz et al., 2014).
4. Figure 2: The reviewer finds the choice of sequences used for the phylogentic analysis strange. Vertebrate α1-, α2- and β-adrenergic receptors and insect β-adrenergic-like OA receptors were used. It is not clear to me why, for example, α1- and α2-adrenergic-like insect OA receptors were excluded. Furthermore, no outgroup was used and the phylogenetic tree was apparently not rooted. The corresponding description in the methods section is very short. For example, it is not clear whether complete receptor sequences were used or whether the sequences were pre-trimmed to exclude highly variable regions, as is commonly done. In the legend, scientific species names should be written in italics.
Linguistic / grammatical inadequacies, typos, inappropriate references etc. (These are examples. The list is certainly not complete.)
Lines 15/16 “Octopamine acts by binding to a specific Octopamine Receptor (OAR)”: Use the plural: “… to specific optopamine receptors.”
Line 25: Put a full stop at the end of the sentence.
Line 25ff “The open reading frame (ORF) of PxOA2B3 was phylogenetically analyzed then its expression levels were determined by quantitative real-time PCR (qPCR).”: Think about writing “The open reading frame (ORF) of PxOA2B3 was phylogenetically analyzed and receptor mRNA its expression levels were determined …”.
Line 29: Delete “)” after “qPCR”.
Lines 44-45 “new insecticides that are both highly selective to target compounds and have low toxicity”: Did you mean “low toxicity to non-target organisms”? I think high toxicity to the target organism would be desirable.
Lines 49-50 “Biogenic amines …”: More suitable references, e.g. current review articles, can be found here.
Lines 54-56 “When in the central nervous system, OA regulates sensory organs such as flight muscles, fat bodies, hemolymph, and fallopian tubes.”: This sentence doesn't make any sense.
Line 57 “thymus of Drosophila”: What do you mean by "thymus of Drosophila"? The thymus is a specialized primary lymphoid organ of the immune system of vertebrates. Until now I didn't know that Drosophila has a thymus. Reference [19] is on autocrine regulation of ecdysone synthesis by a β3-octopamine receptor in the prothoracic gland.
Line 94 “thrice”: The term “trice” is old-fashioned and is rarely used except in a special context. Consider replacing with “three times”. See also line 136.
Lines 102-104 “By querying the P. xylostella transcriptional database with a putative gene sequence (XM_011557721.1), XM_011557721.1 was identified as a candidate gene for the OA2B3 receptor.”: I do not understand that. Was the transcriptome data baited with the same sequence that was eventually found in it?
Line 143 “Construction of stable transformation cell line”: Consider typing “Construction of a cell line stably expressing PxOA2B3”
Line 153: Replace “ug/ml” by “µg/ml”.
Line 177 “OA genes”: Change to “OA receptor genes”.
Lines 198-199 “PxOA2B3 had amino acid sequence similarities of 45.7% with DmOA2B3, TcOA2B3, and AmOA2B3.”: Is it correct that you found the absolutely same amino acid sequence similarity of 45.7% with three different sequences? That's hard to believe.
Lines 212-213 “… are respectively marked with filled circles and triangles.”: Consider changing to “… are marked with filled circles and triangles, respectively.”.
Lines 258-259 “Insect OA2B3s are specifically activated by either octopamine or tyramine, resulting in increased levels of intracellular cAMP[22].”: Additional references should be given, e.g. from the first characterized OA2B3s (or better Octβ3Rs) from Drososphila melanogaster and Apis mellifera.
Lines 266-268 „These were measured using HEK-293 cells overexpressing GFP —no significant changes were found in their intracellular cAMP concentration.“: In the opinion of the reviewer, this sentence makes no sense. What does the green fluorescent protein (GFP) have to do with experiments on the activation of a GPCR by biogenic amines?
Lines 313-314 “Cells in the treated group had fold increases in cAMP when compared to the negative control.”: I am assuming that by “negative control” you mean the “untreated control”. Consider changing the sentence to: “Relative change in [cAMP] is given as a multiple of the value obtained for the untreated control (=1).”.
Line 357ff “… octopamine receptor genes from insects … [26-28]”: Reference [26] refers to a α1-adrenergic receptor of the honey bee Apis mellifera and should be replaced by Balfanz et al. (2014). Reference [28] refers to a β-adrenergic-like OA receptor of the silk moth Bombyx mori. The correct reference for the Nilaparvata lugens receptor would be Wu et al. (2017). Proper reference for Chilo suppressalis receptors would be Wu et al. (2012, 2014). More recently, other β-adrenergic-like OA receptors have been characterized, e.g. for Plutella xylostella (Deng et al., 2021) and Periplaneta americana (Blenau et al., 2022).
Line 370 “… resulting in fold increases of cAMP concentration … ”: Consider typing “… which results in an increase in the intracellular cAMP concentration …”.
Lines 376-377 “PxOA2B3 was most reactive to octopamine but also reacted to tyramine, possibly since tyramine is the biosynthetic precursor of octopamine.”: Yes, tyramine is the precursor of octopamine in biosynthesis. However, the activation of OA receptors by tyramine occurs due to the great structural similarity in the structure of the two phenolamines.
Lines 395-396 “Moreover, amitraz as a formamidine acaricide that mainly controls mites, induces its insecticidal activity …”: Activity against mites and ticks should be referred to as “miticidal activity” or “acaricidal activity”. However, amitraz is also used in veterinary medicine against ectoparasitic insects such as fleas (= insecticidal activity). For agricultural purposes, amitraz is used to control pear psylla and whiteflies (= insecticidal activity).
Line 414, line 415 “prothymus”: What is a “prothymus”? Reference [19] is on autocrine regulation of ecdysone synthesis by a β3-octopamine receptor in the prothoracic gland.
Author Response
Specific comments
Point 1: Generally, the octopamine receptor nomenclature of Evans & Maqueira (2005) should be used: Octopamine receptors can be differentiated in α-adrenergic-like and β-adrenergic-like receptors. The α-adrenergic-like receptors evoke Ca2+ release from intracellular stores (α1) or inhibit adenylyl cyclases (α2) whereas the β-adrenergic-like receptors stimulate adenylyl cyclases and thereby raise [cAMP]i. The classification into OA1 and OA2 receptors has gone out of fashion (see e.g., the following recent reviews: Ozoe 2021; Blenau and Baumann, 2016; Ohta and Ozoe, 2014).
Response 1: Thanks, we have revised it. P2, Line 69 to 74.
(Based on molecular and pharmacological data, insect octopamine receptors can be further grouped as α- or β-adrenergic-like octopamine receptors[7,13-14]. The α-adrenergic-like octopamine receptors induce the release of Ca2+ from intracellular stores (α1) or inhibit adenylyl cyclases (α2), whereas the β-adrenergic-like receptors stimulate adenylyl cyclases and thereby increase the levels of [cAMP]i[15-18].)
Point 2: Line 62 “OARs are divided into two”: Did you mean two receptor subfamilies? There are actually three subfamilies of specific octopamine receptors: α1, α2 and β (see Specific comment 1). The recent discovery of the α2-adrenergic octopamine receptors (e.g., Wu et al., 2014; Qi et al., 2017; Blenau et al., 2020; Nakagawa et al., 2022) is neglected throughout the manuscript.
Response 2:Thanks for your guidance, we have modified the description. P2, Line 69 to 74.
Point 3: Lines 65-66 “β-octopamine receptors are subdivided into OA2B1, OA2B2, and OA2B3 receptors”: The number of subtypes of β-adrenergic-like octopamine receptors present depends on the insect species. For example, honeybees possess four β-adrenergic-like octopamine receptors (Balfanz et al., 2014).
Response 3: Thanks for your guidance, we have identified the description and supplied information as required. P2, Line 74 to 79.
Studies have shown that insects generally have multiple β-adrenergic-like octopamine receptors[19-20]. For example, three additional genes in Drosophila (DmOA2B1 to DmOA2B3) have been characterized that encode β-adrenergic-like octopamine receptors that generate cAMP signals[21-22]. Honeybees (Apis mellifera) possess four β-adrenergic-like octopamine receptors (AmOA2B1 to AmOA2B4)
Point 4: Figure 2: The reviewer finds the choice of sequences used for the phylogentic analysis strange. Vertebrate α1-, α2- and β-adrenergic receptors and insect β-adrenergic-like OA receptors were used. It is not clear to me why, for example, α1- and α2-adrenergic-like insect OA receptors were excluded. Furthermore, no outgroup was used, and the phylogenetic tree was apparently not rooted. The corresponding description in the methods section is very short. For example, it is not clear whether complete receptor sequences were used or whether the sequences were pre-trimmed to exclude highly variable regions, as is commonly done. In the legend, scientific species names should be written in italics.
Response 4: Thank you, we have modified the description. P7, Line 252 to Line 258.
Linguistic / grammatical inadequacies, typos, inappropriate references etc. (These are examples. The list is certainly not complete.)
Point 5: Lines 15/16 “Octopamine acts by binding to a specific Octopamine Receptor (OAR)”:Use the plural: “… to specific optopamine receptors.”
Response 5: Thank you, we have revised it. P1, Line 17 to19.
Point 6: Line 25: Put a full stop at the end of the sentence.
Response 6: Thank you, we have revised it. P1, Line 29.
Point 7: Line 25ff “The open reading frame (ORF) of PxOA2B3 was phylogenetically analyzed then its expression levels were determined by quantitative real-time PCR (qPCR).” Think about writing “The open reading frame (ORF) of PxOA2B3 was phylogenetically analyzed and receptor mRNA its expression levels were determined …”.
Response 7: Thank you, we have modified the description. P1, Line 29 to 31.
Point 8: Line 29: Delete “)” after “qPCR”.
Response 8: Thank you, we have revised it. P1, Line 34.
Point 9: Lines 44-45 “new insecticides that are both highly selective to target compounds and have low toxicity”: Did you mean “low toxicity to non-target organisms”? I think high toxicity to the target organism would be desirable.
Response 9: Thank you, we have revised it. P2, Line 52.
Point 10: Lines 49-50 “Biogenic amines …”: More suitable references, e.g., current review articles, can be found here.
Response 10: Thank you, we have modified the description. P2, Line 56 to 59.
Point 11: Lines 54-56 “When in the central nervous system, OA regulates sensory organs such as flight muscles, fat bodies, hemolymph, and fallopian tubes.”: This sentence doesn't make any sense.
Response 11: Thank you, we have modified the description. P2, Line 63 to 67.
Point 12: Line 75 “thymus of Drosophila”: What do you mean by "thymus of Drosophila"? The thymus is a specialized primary lymphoid organ of the immune system of vertebrates. Until now I didn't know that Drosophila has a thymus. Reference [19] Drosophila is on autocrine regulation of ecdysone synthesis by a β3-octopamine receptor in the prothoracic gland.
Response 12: Thank you, we have revised it. P2, Line 88.
Point 13: Line 94 “thrice”: The term “trice” is old-fashioned and is rarely used except in a special context. Consider replacing with “three times”. See also line 136.
Response 13: Thank you, we have revised it. P3, Line 109、P4, Line 158.
Point 14: Lines 102-104 “By querying the P. xylostella transcriptional database with a putative gene sequence (XM_011557721.1), XM_011557721.1 was identified as a candidate gene for the OA2B3 receptor.”: I do not understand that. Was the transcriptome data baited with the same sequence that was eventually found in it?
Response 14. Thank you, we have modified the description. P3, Line 118 to 123.
Point 15: Line 143 “Construction of stable transformation cell line”: Consider typing “Construction of a cell line stably expressing PxOA2B3”
Response 15: Thank you, we have revised it. P4, Line 166.
Point 16: Line 153: Replace “ug/ml” by “µg/ml”.
Response 16: Thank you, we have revised it. P5, Line 178.
Point 17: Line 177 “OA genes”: Change to “OA receptor genes”.
Response 17: Thank you, we have revised it. P5, Line 204.
Point 18: Lines 198-199 “PxOA2B3 had amino acid sequence similarities of 45.7% with DmOA2B3, TcOA2B3, and AmOA2B3.”: Is it correct that you found the absolutely same amino acid sequence similarity of 45.7% with three different sequences? That's hard to believe.
Response 18: Thank you, we have modified the description. P6, Line 231 to 232.
Point 19: Lines 212-213 “… are respectively marked with filled circles and triangles.”:Consider changing to “… are marked with filled circles and triangles, respectively.”.
Response 19: Thank you, we have revised it. P6, Line 247 to 248.
Point 20: Lines 258-259 “Insect OA2B3s are specifically activated by either octopamine or tyramine, resulting in increased levels of intracellular cAMP[22].”: Additional references should be given, e.g. from the first characterized OA2B3s (or better Octβ3Rs) from Drososphila melanogaster and Apis mellifera.
Response 20: Thank you, we have revised it. P8, Line 303 to 304.
Point 21: Lines 266-268 „These were measured using HEK-293 cells overexpressing GFP —no significant changes were found in their intracellular cAMP concentration.“: In the opinion of the reviewer, this sentence makes no sense. What does the green fluorescent protein (GFP) have to do with experiments on the activation of a GPCR by biogenic amines?
Response 21: Thank you, we have modified the description. P8, Line 314 to 315.
Point 22: Lines 313-314 “Cells in the treated group had fold increases in cAMP when compared to the negative control.”: I am assuming that by “negative control” you mean the “untreated control”. Consider changing the sentence to: “Relative change in [cAMP] is given as a multiple of the value obtained for the untreated control (=1).”.
Response 22: Thank you, we have modified the description. P9, Line 362 to 363.
Point 23: Line 357ff “… octopamine receptor genes from insects … [26-28]”: Reference [26] refers to a α1-adrenergic receptor of the honey bee Apis mellifera and should be replaced by Balfanz et al. (2014). Reference [28] refers to a β-adrenergic-like OA receptor of the silk moth Bombyx mori. The correct reference for the Nilaparvata lugens receptor would be Wu et al. (2017). Proper reference for Chilo suppressalis receptors would be Wu et al. (2012, 2014). More recently, other β-adrenergic-like OA receptors have been characterized, e.g. for Plutella xylostella (Deng et al., 2021) and Periplaneta americana (Blenau et al., 2022).
Response 23: Thank you, we have modified the description. P10, Line 414 to 416.
Point 24: Line 370 “… resulting in fold increases of cAMP concentration …” Consider typing “… which results in an increase in the intracellular cAMP concentration …”.
Response 24: Thank you, we have revised it. P10, Line 428.
Point 25: Lines 376-377 “PxOA2B3 was most reactive to octopamine but also reacted to tyramine, possibly since tyramine is the biosynthetic precursor of octopamine.”: Yes, tyramine is the precursor of octopamine in biosynthesis. However, the activation of OA receptors by tyramine occurs due to the great structural similarity in the structure of the two phenolamines.
Response 25: Thank you, we have modified the description. P10, Line 436.
Point 26: Lines 395-396 “Moreover, amitraz as a formamidine acaricide that mainly controls mites, induces its insecticidal activity …”: Activity against mites and ticks should be referred to as “miticidal activity” or “acaricidal activity”. However, amitraz is also used in veterinary medicine against ectoparasitic insects such as fleas (= insecticidal activity). For agricultural purposes, amitraz is used to control pear psylla and whiteflies (= insecticidal activity).
Response 26: Thank you, we have revised it. P11, Line 457.
Point 27: Line 414, line 415 “prothymus”: What is a “prothymus”? Reference [19] is on autocrine regulation of ecdysone synthesis by a β3-octopamine receptor in the prothoracic gland.
Response 27: Thank you, we have revised it. P12, Line 498 to 499.
Reviewer 2 Report
In this manuscript, the PxOA2B3 of Plutella xylostella was cloned by the authors. Multiple sequence alignments and cluster analysis were performed on PxOA2B3 and other OARs. RT-qPCR was used to analyze the mRNA expression level of PxOA2B3 in different development stages and tissues of the diamondback moth. In addition, the authors examined the effect of several biogenic amines, OA agonists, and antagonists on intracellular cAMP concentrations in HEK-293 cells stably expressing PxOA2B3. Ultimately, it was identified that silencing PxOA2B3 resulted in a shortened pupation time in larvae. Although the authors' study adds to the understanding of the physiological function of OA2B3 in insects, there are many English writing errors in the article. The discussion of the manuscript is insufficient. I think the manuscript is not acceptable in present version.
Specific comments:
1. It is necessary to have a fuller discussion of the expression pattern of PxOA2B3 and the phenotype after RNAi comparing with the literatures. For example, the possible reasons for the significant differential expression of PxOA2B3 in male and female worms, the reasons for choosing 4 tissues of larvae to analyze the expression levels, and how does this gene affect the pupation time?
2. The authors need to add experiments to answer the following questions. Does silencing PxOA2B3 affect larval survival and ecdysone content? Would injection of a PxOA2B3 antagonist lead to a similar phenotype?
3. The authors should confirm the dose of dsRNA used in the RNAi method. In Figure 5 line 345 the authors wrote that 100ng of dsRNA was injected. However, "75ng/larva" is written in method section 2.8, line 184. Beyond that, is this dsRNA concentration too low for lepidopteran RNAi?
4. Line 15 to 16, “Octopamine acts by binding to a specific Octopamine Receptor (OAR)” is clearly a wrong statement.
5. β-adrenergic-like octopamine receptor (line 25) and β3 octopamine receptor (line 47) which is the full name of PxOA2B3?
6. Line 55, “OA regulates sensory organs such as flight muscles, fat bodies…”. Is the author sure that these tissues are sensory organs?
7. Line 89 to 90, "pupal larvae" indicates which developmental stage?
8. The manufacturer and origin of many reagents and instruments in the manuscript are missing, such as RNA extraction kits in line 96, penicillin and streptomycin in line 145.
9. Figure 2: Why not select insect α-adrenergic receptors but human genes for analysis in the phylogenetic tree of PxOA2B3?
10. Figure 3: How did the authors analyze the data?
11. Line 265 to 266, the author wrote that “Treatment with other biogenic amines did not significantly change intracellular cAMP concentrations”, but dopamine also significantly increases cAMP concentrations which showed in Figure 5A.
12. In Section 3.4 the authors need to clarify how many biological replicates were set up in the RNAi assay, and how many test insects were included in each sample?
13. Line 423 to 425, “These findings advance studies on the physiological functions of PxOA2B3 and provide new targets for development of novel insecticides.” If silencing PxOA2B3 only shortens the time it takes to pupate, why do the authors think it could be a new insecticide target?
Author Response
Response to Reviewer 2 Comments
Specific comments:
Point 1: It is necessary to have a fuller discussion of the expression pattern of PxOA2B3 and the phenotype after RNAi comparing with the literatures. For example, the possible reasons for the significant differential expression of PxOA2B3 in male and female worms, the reasons for choosing 4 tissues of larvae to analyze the expression levels, and how does this gene affect the pupation time?
Response 1: Thank you, we have added related discussions. P11, Line 463 to 483 and P12, Line 495 to 503.
Point 2: The authors need to add experiments to answer the following questions. Does silencing PxOA2B3 affect larval survival and ecdysone content? Would injection of a PxOA2B3 antagonist lead to a similar phenotype?
Response 2: Thanks for your guidance, we have added discussion and some descriptions. P9, Line 376 to 379、P12, Line 495 to 503.
(1)Monitoring the phenotypic changes after injection indicated that the survival rate of P. xylostella had no significant difference with the control group. The mean survival rates of the treated and control groups were 89% and 90%, respectively.
(2)First, the experiments on RNAi with Lepidoptera have frequently been found to be difficult to achieve. Second, the focus of this paper is the research on the pharmacological properties of PxOA2B3, and it is only a preliminary exploration of its physiological function. Therefore, we plan to knock out the receptor using a CRISPR/Cas9 technique to study the role of PxOA2B3 in the physiological activity of P. xylostella in more detail.
(3)At present, there is no specific antagonist for PxOA2B3 among the antagonists tested. If antagonists are used for injection. It is difficult to rule out the effects of antagonists on other OA receptors(PxOA2B1 and PxOA2B2).
Point 3: The authors should confirm the dose of dsRNA used in the RNAi method. In Figure 5 line 345 the authors wrote that 100ng of dsRNA was injected. However, "75ng/larva" is written in method section 2.8, line 184. Beyond that, is this dsRNA concentration too low for lepidopteran RNAi?
Response 3: Thanks for helping me point out the error, we have corrected the errors in the new manuscript. P5, Line 210 and P10, Line 401. We verified the original data, the injection volume for this experiment should be 200ng, the dsOA2B3 concentration is 3350ng/μL, and the dsGFP concentration is 3430ng/μL. The injection volume is 60 nL/larva
Point 4: Line 15 to 16, “Octopamine acts by binding to a specific Octopamine Receptor (OAR)” is clearly a wrong statement.
Response 4: Thank you, we have modified the description. P1, Line 18 to 19.
Point 5: β-adrenergic-like octopamine receptor (line 25) and β3 octopamine receptor (line 47) which is the full name of PxOA2B3?
Response 5: Thank you, we have modified the description. P2, Line 54.
Point 6: Line 55, “OA regulates sensory organs such as flight muscles, fat bodies…”. Is the author sure that these tissues are sensory organs?
Response 6: Thank you, we have corrected the errors in the new manuscript. P2, Line 63 to 67.
Point 7: Line 89 to 90, "pupal larvae" indicates which developmental stage?
Response 7: Thank you, we have corrected the mistake. P3, Line 104
Point 8: The manufacturer and origin of many reagents and instruments in the manuscript are missing, such as RNA extraction kits in line 96, penicillin and streptomycin in line 145.
Response 8: Thank you, we have modified the description. P3, Line 112-113、P4, Line 169 to 170、P5, Line 187.
Point 9:Figure 2: Why not select insect α-adrenergic receptors but human genes for analysis in the phylogenetic tree of PxOA2B3?
Response 9: Thanks for your guidance, we have modified the description. P7, Line 252.
Point 10:Figure 3: How did the authors analyze the data?
Response 10: Thank you, we have added the description. P8, Line 298 to 301.
Point 11: Line 265 to 266, the author wrote that “Treatment with other biogenic amines did not significantly change intracellular cAMP concentrations”, but dopamine also significantly increases cAMP concentrations which showed in Figure 5A.
Response 11: Thank you, we have corrected the mistake. P8, Line 308 to 311 and P11, Line 437 to 438.
Point 12: In Section 3.4 the authors need to clarify how many biological replicates were set up in the RNAi assay, and how many test insects were included in each sample?
Response 12: Thank you, we have modified the description. P5, Line 213-216.
Point 13: Line 423 to 425, “These findings advance studies on the physiological functions of PxOA2B3 and provide new targets for development of novel insecticides.” If silencing PxOA2B3 only shortens the time it takes to pupate, why do the authors think it could be a new insecticide target?
Response 13: Thank you, we have modified the description. P12, Line 514 to 515.
Round 2
Reviewer 1 Report
General comments
This is the revised version of a manuscript on the characterization of an octopamine receptor in the diamondback moth Plutella xylostella which is an important pest insect. The PxOA2B3 receptor was heterologously expressed in HEK293 cells and its pharmacological profile was established. In addition, the authors measured the mRNA expression level in different developmental stages and tissues. RNAi experiments were performed to down-regulate receptor expression in order to find possible functions of the receptor protein. The paper has been significantly improved as a result of the revision.
Specific comments
The authors followed my advice and introduced the octopamine receptor nomenclature of Evans & Maqueira (2005). However, this was not done consistently throughout the manuscript. As a result, a mixture of "old nomenclature" (OA1, OA2A, OA2B and OA3 receptors; e.g., Evans PD, 1981; Roeder T, 1992) and "new nomenclature" (α- and β-adrenergic-like OA receptors; Evans & Maqueira, 2005; many other more recent papers) is currently used. The old nomenclature dates from times when the receptors could only be characterized pharmacologically and their molecular identity was still unknown. The receptor characterized in this work is called PxOA2B3, for example, with the “2B” referring to the old nomenclature. The authors should consider renaming the receptor, for example, to PxOctβ3 or PxOAβ3 receptor (= PxOctbeta3 or PxOAbeta3 receptor). This could then be applied consistently to all other receptors mentioned in the manuscript. I ask the authors (and the editor) to consider this suggestion as “not mandatory”.
Linguistic / grammatical inadequacies, typos, etc.
Lines 62/63 “Like other biogenic amines, octopamine (OA) is a multifunctional 62 phenolamine that is only found at high concentrations in invertebrates.”: Consider typing: “Octopamine (OA) is a multifunctional biogenic amine found at high concentrations only in invertebrates.”
Lines 123-125 “…, The specific primers were designed (OA2B3R_XhoI_F and OA2B3R_PstI_R) (Table 1) for use with PCR analysis.”: Consider typing: „…, specific primers (OA2B3R_XhoI_F and OA2B3R_PstI_R) (Table 1) were designed for PCR amplification.“
Lines 148-149 “and human β-adrenergic receptors”: Human receptors were no longer used in the updated version of the phylogenetic tree.
Line 30 “Dopamine”: Don't capitalize.
Lines 371/371 “Pl. xylostella”: Type “P. xylostella”.
Line 470 “Malpighian tubes”: Type “Malpighian tubules”.
Reviewer 2 Report
My concerns have been addressed. Please doublecheck the English for the entire text.
Author Response
Thank you very much for your good suggestions to improve the quality of the manuscript.
